# Parallel Tempering with Lasso for model reduction in systems biology

**Sanjana Gupta**, **Robin E. C. Lee***, **James R. Faeder***

Department of Computational and Systems Biology, University of Pittsburgh School of Medicine, Pittsburgh, Pennsylvania, United States of America

* robinlee@pitt.edu (RECL); faeder@pitt.edu (JRF)

## Abstract

Systems Biology models reveal relationships between signaling inputs and observable molecular or cellular behaviors. The complexity of these models, however, often obscures key elements that regulate emergent properties. We use a Bayesian model reduction approach that combines Parallel Tempering with Lasso regularization to identify minimal subsets of reactions in a signaling network that are sufficient to reproduce experimentally observed data. The Bayesian approach finds distinct reduced models that fit data equivalently. A variant of this approach that uses Lasso to perform selection at the level of reaction modules is applied to the NF-$\kappa$B signaling network to test the necessity of feedback loops for responses to pulsatile and continuous pathway stimulation. Taken together, our results demonstrate that Bayesian parameter estimation combined with regularization can isolate and reveal core motifs sufficient to explain data from complex signaling systems.

## Author summary

Cells respond to diverse environmental cues using complex networks of interacting proteins and other biomolecules. Mathematical and computational models have become invaluable tools to understand these networks and make informed predictions to rationally perturb cell behavior. However, the complexity of detailed models that try to capture all known biochemical elements of signaling networks often makes it difficult to determine the key regulatory elements that are responsible for specific cell behaviors. Here, we present a Bayesian computational approach, PTLasso, to automatically extract minimal subsets of detailed models that are sufficient to explain experimental data. The method simultaneously calibrates and reduces models, and the Bayesian approach samples globally, allowing us to find alternate mechanistic explanations for the data if present. We demonstrate the method on both synthetic and real biological data and show that PTLasso is an effective method to isolate distinct parts of a larger signaling model that are sufficient for specific data.

**Data Availability Statement:** All code used to generate the results shown in the figures in the paper is available on Github (https://github.com/RuleWorld/SupplementalMaterials/tree/master/Gupta2019; DOI: 10.5281/zenodo.3668765).

**Funding:** This work was funded by NIH grant R35-GM119462 to RECL, and by JRF via the NIGMS-funded (P41-GM103712) National Center for Multiscale Modeling of Biological Systems (MMBioS). The funders had no role in study design, data collection and analysis, decision to publish, or preparation of the manuscript.

**Competing interests:** The authors have declared that no competing interests exist.

This is a *PLOS Computational Biology* Methods paper.

## Introduction

Cells use complex networks of proteins and other biomolecules to translate environmental cues into various cell fate decisions. Mathematical and computational models are increasingly used to analyze the nonlinear dynamics of these complex biochemical signaling systems [1–4]. As our knowledge of the biochemical processes in a cell increases, reaction network models of cell signaling have been growing more detailed [4–6]. Detailed models are a useful summary of knowledge about a system but they suffer from several drawbacks. First, the complexity may obscure simpler motifs that govern emergent cellular functions [7–9]. Second, the large number of parameters creates a high-dimensional search problem for parameter values where the model fits the data. To mitigate these problems, it is useful to reduce the number of reactions in a model, provided that the reduced model is still able to reproduce a given set of experimental observations. In this work we pose model reduction as a constrained Bayesian parameter estimation (BPE) problem to simultaneously calibrate and reduce models. Given a prior reaction network model, our method finds minimal subsets of non-zero parameters that fit the data.

A number of previous studies have addressed model reduction for biochemical systems, as reviewed in [10]. Some examples include reduction by topological modifications to resolve non-identifiability in models [11, 12] and reduction by timescale partitioning [13–15]. Non-identifiability arises when multiple unique parameterizations of a model give the same model output. Quaiser et al. [11] and Maiwald et al. [12] developed methods to find non-identifiable parameters and used this analysis to resolve non-identifiability by model simplifications such as lumping or removal of reactions. The simplification step, however, is not automated and requires a skilled modeler. Timescale partitioning methods use timescale separations in the reaction kinetics to apply model reduction based on quasi-steady-state and related approximations [10, 14]. Both of these methods generate reduced models but do not carry out parameter estimation to fit experimental data. Gabel et al. [16] recently developed FaMoS (Flexible and dynamic Algorithm for Model Selection), a method that uses heuristic search algorithms to search the space of submodel topologies within a larger model. However, each proposed submodel has to be individually refit to experimental data, and the heuristic search algorithms used are not guaranteed to return all possible submodel topologies that fit the data. Maurya et al. [17] used mixed-integer nonlinear optimization to combine parameter estimation with model reduction by reaction elimination, a technique common in the field of chemical engineering [18, 19]. This approach requires an additional binary parameter for every reaction in the model, and the genetic algorithms used for the optimization only provide point estimates of the parameters.

Here, we develop reaction elimination in a Bayesian framework that combines parameter estimation and model reduction without requiring additional parameters. BPE can be used to characterize high-dimensional, rugged, multimodal parameter landscapes common to systems biology models [1, 20–23] but suffers from the drawback that the Markov Chain Monte Carlo (MCMC) methods commonly used to sample model parameter space are often slow to converge and do not scale well with the number of model parameters. We recently showed that Parallel Tempering (PT), a physics-based method for accelerating MCMC [24], outperforms conventional MCMC for systems biology models with up to dozens of parameters [20]. Here, we apply Lasso (also known as L1 regularization), a penalty on the absolute values of the parameters being optimized, to carry out model reduction. In the fields of statistics and machine learning, Lasso is widely used for variable selection to identify a parsimonious model

—a minimal subset of variables required to explain the data [25]. In the context of biology, Lasso has been widely applied to gene expression and genomic data, typically in combination with standard regression techniques [26–30] and less commonly in Bayesian frameworks [31, 32]. In the mechanistic modeling context, Lasso regression has been used to predict cell type specific parameters in ODE reaction network models [33], but to our knowledge it has not been implemented to reduce such models.

Our method, PTLasso, combines PT with Lasso regularization to simultaneously calibrate and reduce models. The core idea is that every reaction in the model is governed by a rate constant parameter that, when estimated as zero, removes the reaction from the model simulation. Since the approach is Bayesian, PTLasso can extract multiple minimal subsets of reactions if present, which provides alternate mechanisms to explain the data. We use synthetic data to demonstrate that PTLasso is an effective approach for model reduction. We also use PT with Lasso on groups of parameters (grouped Lasso) with real biological data in a larger model of NF-$\kappa$B signaling to select over reaction-network modules instead of individual reactions. Grouped Lasso can test mechanistic hypotheses about the necessity of signaling modules, such as feedback loops, to explain data from particular experimental conditions. Overall, our results demonstrate that BPE combined with regularization is a powerful approach to dissect complex systems biology models and identify core reactions that govern cell behavior.

The remainder of this paper is organized as follows. In Methods we provide an overview of the PTLasso approach with in-depth descriptions of PT, regularization with Lasso and grouped Lasso, and the setup of computational experiments. In Results we demonstrate PTLasso on synthetic examples of increasing complexity followed by an application of the grouped Lasso approach to address mechanistic questions in NF-$\kappa$B signaling. Finally, in Discussion we highlight advances as well as limitations of the method and present the implications of this study for the broader context of biological modeling and analysis.

## Methods

In this work we use Bayesian parameter estimation (BPE) for model reduction. Here, we present an overview of BPE using the Metropolis-Hastings (MH) and PT algorithms for MCMC sampling. Our presentation of these algorithms is modified from Gupta et al. [20]. Following this, we describe the application of regularization in the context of MCMC sampling using either Lasso or grouped Lasso. Finally, we describe the basic steps of the computational experiments, including generation of the synthetic data for fitting, choosing the starting parameter configurations for the MCMC chains, convergence testing and selection of hyperparameters.

Following [20], BPE methods aim to estimate the probability distribution for the model parameters conditioned on the data. The probability of observing the parameter vector, $\vec{\theta}$, given the data, $Y$, is given by Bayes' rule

$$p(\vec{\theta}|Y) \propto p(Y|\vec{\theta})p(\vec{\theta}). \tag{1}$$

Here, $p(Y|\vec{\theta})$ is the conditional probability of $Y$ given $\vec{\theta}$, and is described by a *likelihood model*. For the ordinary differential equation (ODE) models in this study, we assumed Gaussian experimental measurement error, in which case the likelihood of a parameter vector, $\vec{\theta}$, is given by

$$L(\vec{\theta}) = e^{-\Sigma_S \Sigma_T (Y_{\text{sim}}(\vec{\theta}) - Y_{\text{expt}})^2 / 2\sigma^2}, \tag{2}$$

where $S$ is a list of the observed species, $T$ is a list of the time points at which observations are made, $\sigma$ is the standard deviation of the likelihood model and can be different for different

species and time points, $Y_{\text{sim}}(\vec{\theta})$ is the model output for parameter vector $\vec{\theta}$, and $Y_{\text{expt}}$ is the corresponding experimental data. $p(Y|\vec{\theta})$ is equal to the normalized $L(\vec{\theta})$. $p(\vec{\theta})$ is the independent probability of $\vec{\theta}$, often referred to as the *prior distribution*, which represents our prior beliefs about the model parameters. It can be used to restrict parameters to a range of values or even to limit the number of nonzero parameters, as discussed further below.

## MCMC sampling

MCMC methods sample from the posterior distribution, $p(\theta|Y)$, by constructing a Markov chain with $p(\theta|Y)$ as its stationary distribution. Following the notation of Metropolis *et al.* [34], we define the energy of a parameter vector $\vec{\theta}$ as

$$E(\vec{\theta}) = -\log L(\vec{\theta}) - \log p(\vec{\theta}), \tag{3}$$

where $L$ and $p$ are the likelihood and prior distribution functions defined above. In this section we will briefly describe the Metropolis-Hastings and Parallel Tempering algorithms for MCMC sampling.

**Metropolis-Hastings algorithm.** The Metropolis-Hastings (MH) algorithm is a commonly-used MCMC algorithm for BPE [35]. At each step, $n$, the method uses a proposal function to generate a new parameter vector, $\vec{\theta}_{\text{new}}$, given the current parameter vector, $\vec{\theta}_n$. A common choice of proposal function is a normal distribution centered at $\vec{\theta}_n$:

$$f(\vec{\theta}_{\text{new}}, \vec{\theta}_n) \propto e^{-\alpha|\vec{\theta}_{\text{new}} - \vec{\theta}_n|^2}. \tag{4}$$

For any $f$ that is symmetric with respect to $\vec{\theta}_{\text{new}}$ and $\vec{\theta}_n$, the move $\vec{\theta}_{n+1} = \vec{\theta}_{\text{new}}$ is accepted with probability $\min(1, e^{-\Delta E})$, where $\Delta E = E(\vec{\theta}_{n+1}) - E(\vec{\theta}_n)$. If the move is not accepted $\vec{\theta}_{n+1}$ is set to $\vec{\theta}_n$.

**Parallel tempering.** In PT (also referred to as replica exchange Monte Carlo [24]), several Markov chains are constructed in parallel, each with a different temperature parameter, $\beta$, which scales the acceptance probability from the MH algorithm, which is now given by $\min(1, e^{-\beta\Delta E})$. A Markov chain with $\beta = 1$ samples the true energy landscape as in MH. Higher temperature chains have $\beta < 1$ and accept unfavorable moves with a higher probability, sampling parameter space more broadly. Tempering refers to periodic attempts to swap parameter configurations between high and low temperature chains. These moves allow the low temperature chain to escape from local minima and improve both convergence and sampling efficiency [20, 24]. Following [20], the PT algorithm is as follows:

1. For each of $N$ swap attempts (called "swaps" for short)

   a. For each of $N_c$ chains (these can be run in parallel)

      i. Run $N_{\text{MCMC}}$ MH steps

      ii. Record the values of the parameters and energy on the final step.

   b. For each consecutive pair in the set of chains in decreasing order of temperature, accept swaps with probability $\min(1, e^{\Delta\beta\Delta E})$, where $\Delta E$ and $\Delta\beta$ are the differences in the energy and $\beta$ of the chains, respectively.

Note that in the Results we often refer to a parameter vector obtained from the lowest temperature chain at a particular swap as a sample. Ensemble fits are shown by subsampling parameter vectors from the lowest temperature chain.

Adapting the step size and the temperature parameter can further increase the efficiency of sampling [24], but varying parameters during the construction of the chain violates the assumption of a symmetric proposal function (also referred to as "detailed balance"). It is therefore advisable to do this during a "burn-in" phase prior to sampling. Another way to increase efficiency of sampling for parameters that may be on different scales is to sample in log-space as we have done for all of the examples in this manuscript.

## Regularization with Lasso

Lasso regularization penalizes the L1-norm (sum of absolute values) of the parameter vector, which biases all model parameters towards a value of zero [25]. In a Bayesian framework, the Lasso penalty is equivalent to assuming a Laplace prior on each parameter $\theta_i$ given by

$$p(\theta_i) = \frac{1}{2b} \exp\left(-\frac{|\theta_i - \mu|}{b}\right), \tag{5}$$

where $b$ is the width and $\mu$ is the mean, which is set to zero for variable selection in linear parameter space. The energy function is then

$$E(\vec{\theta}) = -\log L(\vec{\theta}) + \sum_{i=1}^{n_{\text{par}}} \frac{|\theta_i - \mu|}{b}, \tag{6}$$

where $n_{\text{par}}$ is the number of model parameters, and we have dropped the constant term arising from the normalization constant $\frac{1}{2b}$ in Eq 5. From Eq 6 it can be seen that the regularization strength is inversely proportional to $b$. For efficiency we perform parameter estimation in log parameter space, so instead of regularizing by setting $\mu$ to zero, we set it to a large negative value, such that the parameter value is small enough that it does not affect the dynamics of the model variables on the timescale of the simulation.

## Regularization with grouped Lasso

To account for modularity in complex signaling networks [36], we use grouped Lasso, to perform selection at the level of reaction modules instead of individual reactions (Note that this differs from the standard Group Lasso penalty [37] that is typically used for regression problems). All reactions in a module share a common penalty parameter that is multiplied with a reaction-specific parameter to get the full reaction rate constant.

For every reaction $i$ in module $m$, the reaction rate constant is given by

$$\theta_i = k_i \lambda_m, \tag{7}$$

where $\lambda_m$ is the penalty parameter for module $m$ and $k_i$ is a reaction-specific parameter. Defining $\theta_i' = \log(\theta_i)$, $k_i' = \log(k_i)$, and $\lambda_m' = \log(\lambda_m)$, we have

$$\theta_i' = k_i' + \lambda_m'. \tag{8}$$

The energy function is then

$$E(\vec{\theta'}) = -\log L(\vec{\theta'}) + \sum_{m=1}^{n_{\text{mod}}} \frac{|\lambda_m' - \mu|}{b} + \sum_{i=1}^{n_{\text{par}}} f(k_i'), \tag{9}$$

where

$$f(k_i') = \begin{cases} 0, & \text{if } k_i' \in (LB_i, UB_i) \\ \infty, & \text{otherwise.} \end{cases} \tag{10}$$

Here, $n_{\mathrm{mod}}$ is the number of modules and $LB_i$ and $UB_i$ are parameters that restrict the reaction-specific parameters. $UB_i$ is chosen such that when $\lambda_m'$ is within the Laplace prior boundaries, i.e., $\lambda_m' \approx \mu$, the maximum value of $\theta_i', \approx UB_i + \mu$, is small enough that it does not affect the dynamics of the model variables on the timescale of the simulation. For the application to NF-$\kappa$B signaling we chose $\mu = -25$, $LB_i = -5$ and $UB_i = 10$ for all $i$.

## Synthetic data sets used in model calibration

For the two examples presented in Results that used synthetic data, we generated the sets labeled "true data" by simulating the model with a single set of parameter values (labeled as "true parameter values") and sampling with a fine time resolution. We then generated 10 noisy replicates of this data at a coarser set of time points by adding Gaussian noise with mean of zero and variance of either 10% or 30% of the true value at each point. The mean and variance of the replicates then defined the "observed data" used for fitting.

## Constraining the model

We use two kinds of constraints in fitting, soft constraints and hard constraints. Soft constraints can be violated, but are associated with a finite penalty [38]. For example, the energy function penalizes parameter vectors for producing model outputs that deviate from the data. Hard constraints, on the other hand, cannot be violated because they are associated with an infinite penalty. We used hard constraints in the NF-$\kappa$B signaling model to enforce certain known properties of the NF-$\kappa$B system, such as that the exit rate of NF-$\kappa$B-I$\kappa$B complex from the nucleus is greater than that of free NF-$\kappa$B [39]. A full list of constraints applied to the NF-$\kappa$B signaling model is listed in S1 Table.

## MCMC chain initialization

All MCMC chains must be initialized with a starting parameter vector. For simple examples, such as the pulse-generator motif and linear dose-response models, chains were initialized by randomly sampling from the prior until a parameter vector with energy below a threshold is found. For more complex examples, to avoid long burn-in periods when starting from unfavorable start points, parameter vectors obtained from PT (or PTLasso) chains that were previously run with similar data or hyperparameter configurations were used to initialize the current PT (or PTLasso) chains. For example, parameter vectors obtained for one NF-$\kappa$B trajectory could be used as a start point for fitting a different NF-$\kappa$B trajectory, or a PTLasso chain with a small value of $b$ (more constrained) could be initialized from a parameter set obtained from PTLasso with a large value of $b$ (less constrained). The exact procedures used to generate the starting configurations used in all computational experiments are provided in the Supplemental Code available at https://github.com/RuleWorld/SupplementalMaterials/tree/master/Gupta2019.

## Convergence testing

To check for convergence, PT (or PTLasso) was run twice for each computational experiment, and the two parameter chains were used to calculate the Potential Scale Reduction Factor (PSRF) for each model parameter (S2 Table). The PSRF compares intra-chain and inter-chain

variances for model parameter distributions and serves as a measure of convergence [40]. In keeping with the literature, we consider a PSRF less than 1.2 [21, 22, 40] as consistent with convergence. We also calculated the stricter Multivariate PSRF (MPSRF), which extends PSRF by checking for convergence of parameter covariation (S3 Table). Third-party MATLAB libraries used for the PSRF and MPSRF calculations are available at https://research.cs.aalto.fi/pml/software/mcmcdiag/.

For models with a large number of parameters, such as the 26-parameter NF-$\kappa$B signaling model, the number of PT (or PTLasso) swaps needed for convergence was large and time consuming to obtain in a single run. Instead of running two long PT (or PTLasso) chains each of length $N$, we picked two favorable initial conditions and from each ran a set of $M$ PT (or PTLasso) chains of length $N/M$ in parallel to reduce wall clock time. We calculated the univariate PSRF of the $M$ energy chains within each group, and if PSRF was less than 1.2, we assumed that the chains were sampling the same energy basin and combined them (S4 and S5 Tables). This gave us two groups of $N$ PT (or PTLasso) samples that we used to test convergence of parameter distributions.

PSRF and MPSRF values for each computational experiment are shown in S2 and S3 Tables respectively. We also show in S6 Table that the step acceptance rates for most chains are close to the optimal value of 0.234 [41]. S7 Table shows the swap acceptance rates of the two lowest temperature chains for each computational experiment.

### Hyperparameter selection

The hyperparameters associated with PTLasso are $\mu$ and $b$, the mean and width of the Laplace prior on each parameter that is being regularized. For simplicity, we keep these the same for all model parameters, although they could in principle vary, which would lead to a more difficult inference problem. To select the hyperparameters, we varied $b$ and used the "elbow" in the negative log likelihood vs. $b$ plot to find the smallest value of $b$ (maximum regularization strength) that does not substantially increase the negative log likelihood of the fit [20, 42]. We also checked that the results were insensitive to small variations in $\mu$ (S1B, S2B, S4B and S4C Figs).

For more computationally expensive models, we used hyperparameter estimates close to those obtained from the smaller synthetic models and compared the average log likelihoods of the fits from PT and PTLasso. For all of the examples shown, we found that the fit with PTLasso is at least as good as the fit with PT (Fig 4E, S1C, S2C, S4D and S4E Figs).

### Software

All results reported in this work were obtained using `ptempest` [20], which is a MATLAB package for parameter estimation that implements PT with support for regularization. `ptempest` uses MATLAB's Mex interface to support the efficient integration of ODEs in C using the CVODE library and is directly compatible with the popular rule-based modeling software BioNetGen [43] which enables use with models built in both Systems Biology Markup Language (SBML) [44] and the BioNetGen Language (BNGL). The source code is available at http://github.com/RuleWorld/ptempest.

## Results

### Reduced motifs can be inferred from dense reaction-networks in the absence of a prior architecture

To demonstrate that PTLasso can recover a minimal model architecture without prior knowledge of the reaction network, we used synthetic time-course data to infer a pulse-generator

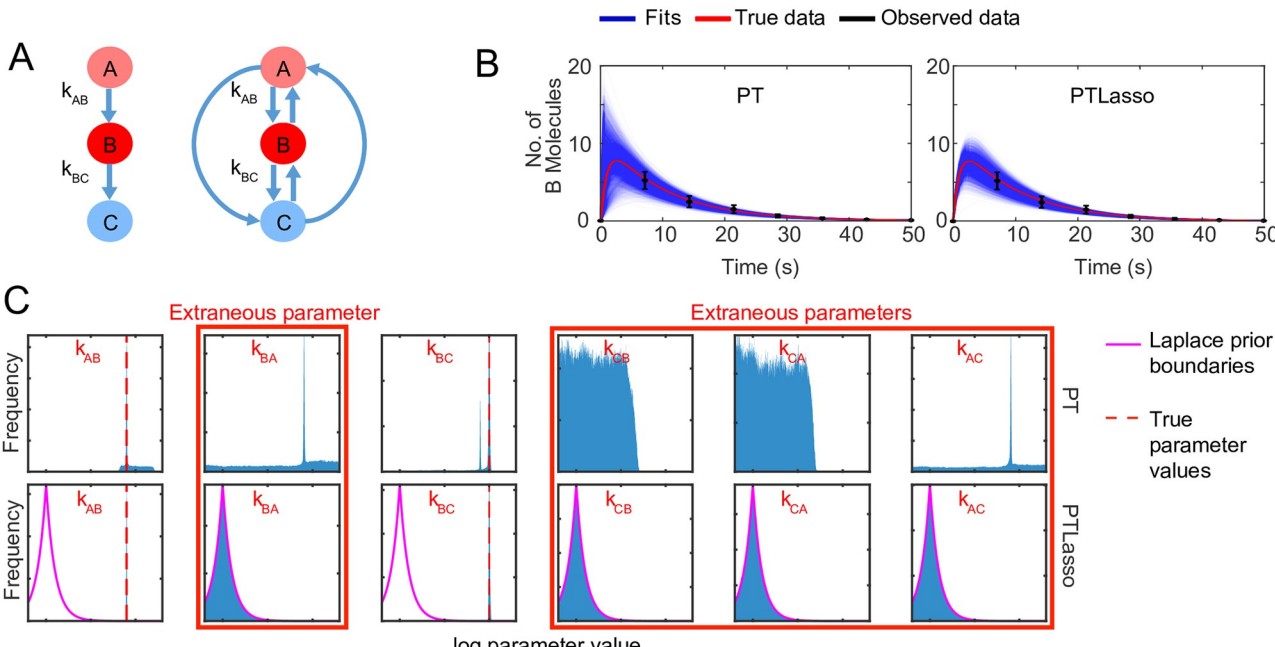

**Fig 1. Model reduction using PTLasso with a fully connected 3-node graph. A)** Motif used to generate the observed data (left) and reaction network diagram of the fully connected 3-node network used as the starting point for PTLasso (right). The initial concentration of A (light red) is 100 molecules. The initial concentrations of B and C are 0. The concentration of B (red) is observed at multiple time points, but the concentration of C (blue) is not observed. Each reaction has an associated rate constant parameter. $k_{AB} = 0.1s^{-1}$ and $k_{BC} = 1s^{-1}$ are the true parameter values (rate constants not specified were set to zero). **B)** Fits of the model to the data with PT (left) and PTLasso (right). Transparent blue lines show ensemble fits (from 4,000 parameter samples, 100 time points per trajectory), red line shows the true data (100 time points), and the black error bars show the mean ± standard deviation of the observed data (8 time points). **C)** Frequency histograms showing probability distributions of the parameters (from 400,000 parameter samples) for fits with PT (top row) and PTLasso (bottom row). The range of log parameter values on each x-axis is −12 to 3, which covers the full range over which parameters were allowed to vary. The y-axis of each panel is scaled to the maximum value of the corresponding distribution to emphasize differences in shape. The pink lines show the boundaries of the Laplace prior with $\mu = -10$, $b = 1$ and the dashed red lines in panels for $k_{AB}$ and $k_{BC}$ show the true parameter values. A parameter distribution confined within the Laplace prior boundaries indicates that the parameter is extraneous (panels with red border).

motif from a fully connected 3-node network of unimolecular reactions (S8 Table). The motif A→B→C (Fig 1A, left), modeled as a system of ODE's, was used to generate a time course for species B after initializing the system with 100 molecules of species A at time $t = 0$ (red curves in Fig 1B labeled "true data"). As described in Methods, Gaussian noise (mean = 0, standard deviation = 30% of the true data value) was added to generate ten noisy trajectories that were sampled at eight time points (S1A Fig) to simulate the effects of experimental noise and cell-to-cell variability. The mean and standard deviation of these synthetic trajectories formed the "observed data" (black points and error bars in Fig 1B) used for subsequent parameter estimation and model reduction.

PT and PTLasso were then used to fit this data using the fully-connected 3-node network comprised of six reactions (Fig 1A, right). Time courses from PT and PTLasso (Fig 1B) both fit the observed data (S1C Fig), but the PTLasso fits are more similar to the true data at times before the first observed data point. PT finds parameter probability distributions (Fig 1C, top row) that exhibit sharp peaks near the true values of the two nonzero parameters that were used to generate the data, $k_{AB}$ and $k_{BC}$, but finds significant probability for other values of these parameters and non-zero values for the other rate constants in the complete network that should have zero value (labeled "extraneous"). By contrast, PTLasso (Fig 1C, bottom row) recovers tight distributions near the true values of the two nonzero parameters that lie well

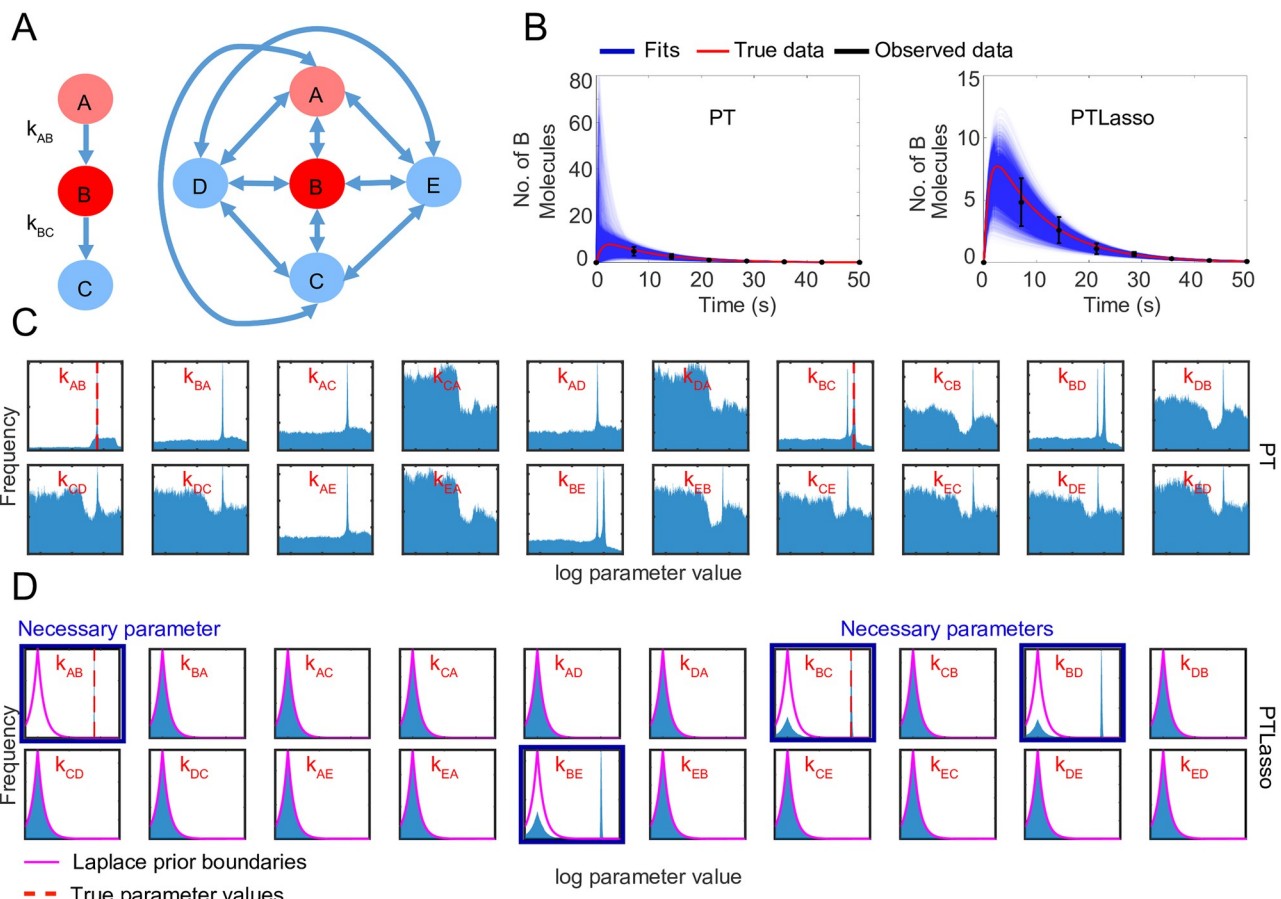

**Fig 2. Model reduction using PTLasso with a fully connected 5-node graph. A)** Motif used to generate the observed data (left) and reaction network diagram of the fully connected 5-node network used as the starting point for PTLasso (right). $k_{AB} = 0.1s^{-1}$ and $k_{BC} = 1s^{-1}$ are the true parameter values (rate constants not specified were set to zero). The initial concentration of A (light red) was set to 100 molecules, while the initial concentrations of B, C, D and E were set to 0. The concentration of B (red) is observed at multiple time points, but the concentrations of C, D and E (blue) are not observed. **B)** Fit of the model to the data with PT (left) and PTLasso (right). Transparent blue lines show ensemble fits (from 7,000 parameter samples, 100 time points per trajectory), red line shows the true data (100 time points), and the black error bars show the mean ± standard deviation of the observed data (8 time points). **C)** Frequency histograms showing probability distributions of the parameters (from 700,000 parameter samples) for fits with PT and **D)** PTLasso. The range of log parameter values on each x-axis is −12 to 3, which covers the full range over which parameters were allowed to vary. The y-axis of each panel is scaled to the maximum value of the corresponding distribution to emphasize differences in shape. The pink lines show the boundaries of the Laplace prior with $\mu = -10$, $b = 1$ and the dashed red lines in panels for $k_{AB}$ and $k_{BC}$ show the true parameter values. A parameter distribution that deviates from the prior is necessary (panels with blue border).

outside the Laplace prior, while the probability distributions for the extraneous parameters all conform tightly to the prior distribution, indicating that the corresponding reactions can be removed from the network. Taken together, these results demonstrate that PTLasso can recover network architecture and parameter values that are not inferred by PT alone.

To determine if the method scales to larger networks, we applied PT and PTLasso to a fully connected 5-node network (S9 Table, Fig 2A, S2A Fig). As with the 3-node example, PTLasso fits for a complete 5-node network are more similar to the true data than fits with PT alone (Fig 2B). Similarly, rate constant parameter distributions with PT are all broad (Fig 2C), whereas the extraneous parameters for the PTLasso fits were within the Laplace prior (Fig 2D). In addition to a tight distribution near the true value for $k_{AB}$, PTLasso recovered bimodal distributions for $k_{BC}$, $k_{BD}$, and $k_{BE}$, suggesting that the essentiality of each of the reactions B→C, B→D and B→E depends on which of the other two are included. This is because the model

A→B→C is indistinguishable from A→B→D and A→B→E without more information about the system. Even though the marginal posterior distributions show all three parameters playing a role, parameter covariation (S2D Fig) reveals that only one of the reactions B→C, B→D, B→E is simultaneously active and rate constant distributions for the other two are centered at $10^{-10}$ (proxy for 0 when sampling in log-scale). The same covariation plot obtained without Lasso does not show similar clustering (S2E Fig). Taken together, these results show that PTLasso correctly identifies network parameters and suggests that A→B→C, A→B→D, and A→B→E are alternate reduced models for the data.

The primary noise model that we have chosen for all the synthetic experiments in this paper is Gaussian noise added to the data. However, one might ask how the results of PTLasso are affected when noise is added to the true parameter values instead, which is perhaps a more accurate representation of cell-to-cell variability in biological signaling systems. To test this, we perturbed the true parameters (log $k_{AB}$ = −1, log $k_{BC}$ = 0) 10 times with Gaussian noise (mean = 0, standard deviation = 0.05) (S3A Fig). The mean and standard deviation of the resulting 10 noisy model outputs formed the observed data for fitting (S3B Fig). The results of PTLasso were qualitatively the same when fitting fully connected three-node and five-node networks to this data (S3C–S3F Fig).

Overall, these results show that PTLasso is a global approach that can extract correct parameter estimates and architectures of alternate reduced models that fit the data from fully connected networks of varying sizes. This is especially useful in the context of complex cell signaling systems that often have redundant elements, in which case the method can be used to identify alternate signaling mechanisms that fit the data.

## Motifs with specific dose-response relationships can be inferred from a prior network

In the previous section we assumed no prior knowledge of a reaction-network and fitted a simple model output. To demonstrate the extraction of motifs with more complex behaviors in the more likely scenario where there is some prior network of hypothesized molecular interactions, we used PTLasso to extract subnetworks required to produce specific dose-response relationships.

Tyson et al. [9] previously described two simple biochemical models that individually produce linear or perfectly adapting dose-response relationships. We constructed a prior network of a signal, *S*, response, *R*, and intermediate, *X*, by combining the linear and adaptive dose-response models into a single six-parameter model (S10 Table, Fig 3A). We show that PTLasso correctly identifies the linear and adaptive submodels when the combined model is fit to different simulated data. The linear dose-response submodel was used to generate synthetic time courses for *R* in response to increasing levels of *S* (Fig 3B, top row). As earlier, Gaussian noise (mean = 0, standard deviation = 10% of the true data value) was added to each trajectory to simulate experimental noise and cell-to-cell variability, and the mean and standard deviation for each time course was calculated at four distinct time points (including *t* = 0), creating 16 data points that constitute the observed data. As in the previous example of fully connected networks, PTLasso fits of the prior network to the observed data are more similar to the true data than fits from PT alone (Fig 3B, top row). PTLasso recovers tight distributions for $k_{s-rs}$ and $k_{r-0}$, which are the only model parameter values that lie outside the Laplace prior, providing a reduced two-parameter model that is sufficient to produce the synthetic data (Fig 3C and 3D).

When the perfectly adaptive dose-response submodel was similarly used to generate observed data in response to two successive increasing values of *S*, PTLasso reduced the prior

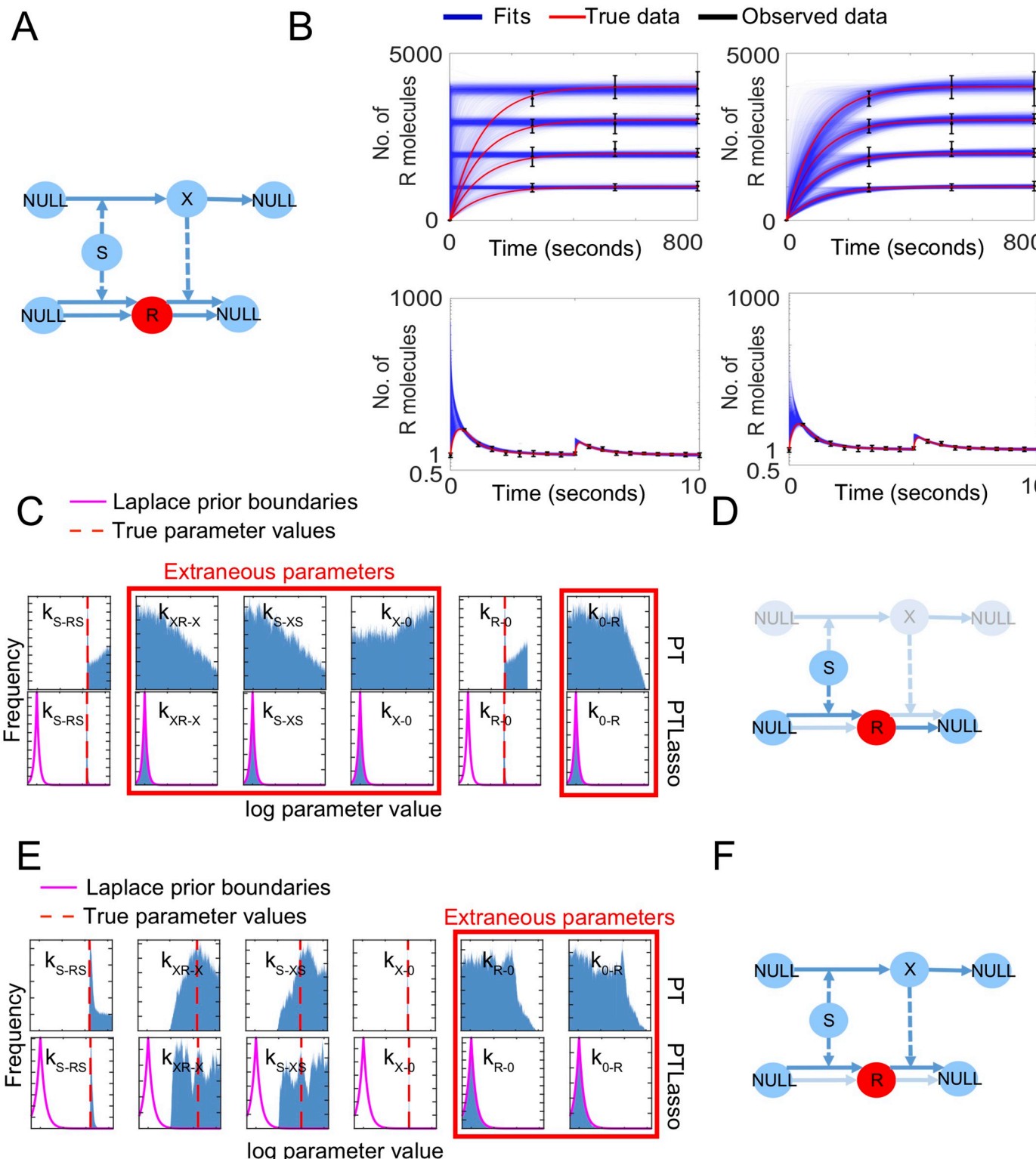

**Fig 3. Motif inference from a prior network constrained with dose-response data. A)** Reaction network diagram of the prior network. The value of the signal *S* is known, response *R* (red) is observed at multiple time points, but intermediate *X* (blue) is not observed. Solid lines show species conversions and dashed lines show influences, where a species affects the rate of the corresponding reaction without being consumed. **B)** Fit of the model to the linear dose-response data (top row, linear scale y-axis) and perfectly adapting dose-response data (bottom row, log scale y-axis) with PT (left) and PTLasso (right). $k_{s-rs} = 10$ s$^{-1}$, $k_{r-0} = 0.01$ s$^{-1}$ are the true parameter values for the linear dose-response data and $k_{s-rs} = 10$ s$^{-1}$, $k_{xr-x} = 10$ molecule$^{-1}$ s$^{-1}$, $k_{s-xs} = 1$ s$^{-1}$, $k_{x-0} = 1$ s$^{-1}$ are the true parameter values for the

perfectly adapting dose-response data (rate constants not specified were set to zero in each case). Transparent blue lines show ensemble fits (from 4,000 parameter samples with 1,000 time points per trajectory for linear dose-response, and from 8,000 parameter samples with 2,000 time points per trajectory for perfectly adapting dose response), red lines show the true data (1,000 time points for linear dose-response, 2,000 time points for perfectly adapting dose response), and the black error bars show the mean ± standard deviation of the observed data. The four increasing linear dose-response values correspond to $S$ values of 1, 2, 3 and 4, and the two successive perfectly adapting dose response responses corresponding to $S$ values of 1 and 2. **C)** Frequency histograms showing probability distributions of the parameters for linear dose response fits (from 400,000 parameter samples) with PT (top) and PTLasso (bottom). **D)** Reduced model corresponding to linear dose-response highlighted in prior network. Faded nodes and arrows are extraneous and are removed from the model. **E)** Frequency histograms showing probability distributions of the parameters for perfectly adapting dose-response fits (from 800,000 parameter samples) with PT (top) and PTLasso (bottom). For panels C and E, the range of log parameter values on each x-axis is −12 to 6, which covers the full range over which parameters were allowed to vary. The y-axis of each panel is scaled to the maximum value of the corresponding distribution to emphasize differences in shape. The pink lines show the boundaries of the Laplace prior with $\mu = -10$, $b = 0.5$ for the linear dose-response model, and $\mu = -10$, $b = 1$ for the perfectly adapting dose-response model, and the dashed red lines show the true parameter values. A parameter distribution confined within the Laplace prior boundaries indicates that the parameter is extraneous (panels with red border). **F)** Reduced model corresponding to perfectly adapting dose-response highlighted in prior network. Faded nodes and arrows are extraneous and are removed from the model.

network to a four-parameter model (Fig 3E and 3F) that fits the data (Fig 3B, bottom row). In this case, parameters $k_{s-xs}$ and $k_{xr-x}$ in the reduced model have broad distributions and are unidentifiable (Fig 3E, bottom row), but PTLasso captures their linear correlation (S4A Fig), which may provide further avenues for model reduction [12]. While signaling systems are complex and can involve large numbers of reactions, not every reaction is relevant for every function. Taken together our results demonstrate that distinct elements of a large reaction-network may be responsible for different complex behaviors and can be successfully isolated using PTLasso.

## A reduced model of NF-$\kappa$B signaling without A20 feedback explains single-cell NF-$\kappa$B responses to a short TNF pulse

Complex biological signaling networks are frequently modular [45, 46] with distinct motifs such as feedback loops that operate on separate time scales [36]. To account for the modular structure of signaling we extended our Lasso approach to grouped Lasso, a technique that applies a module-specific Lasso penalty to all reactions within a particular module (see Methods). PT combined with grouped Lasso finds minimal sets of reaction modules that explain experimental data. We used this method to test the requirement of A20 feedback to explain previously published single-cell NF-$\kappa$B responses to a short TNF pulse [47]. A prior model of NF-$\kappa$B signaling was created by combining simplified elements of models from [39] and [2] (S11 Table). The network was divided into three biologically motivated network modules (Fig 4A). The I$\kappa$B and A20 modules describe negative feedback mediated by the inhibitor I$\kappa$B and negative regulator A20, respectively. The activation module includes all remaining reactions that describe the path from TNF binding to its cognate TNF-receptor (TNFR) to the eventual translocation of NF-$\kappa$B into the nucleus. The reaction rate constants within a module are constrained by a common Lasso penalty parameter (see Methods). If the penalty parameter for a module is estimated as 0, (here, $10^{-25}$ is used as a proxy for 0 when sampling in log scale), the entire module is removed from the simulation. To test which of the three modules is necessary to explain NF-$\kappa$B responses to a single TNF pulse, PTLasso was used to fit the model to three previously published, experimentally obtained, single-cell NF-$\kappa$B responses (Fig 4B) [47]. In addition to the NF-$\kappa$B data, other constraints were applied to make the system behave consistently with known biology. These constraints are listed in S1 Table, and S5 Fig demonstrates that PTLasso correctly followed the imposed parameter covariation.

The probability distributions for the module penalty parameters (Fig 4C) show that the A20 parameter is confined within the prior boundaries while the others have deviated, suggesting that to fit these particular single-cell NF-$\kappa$B trajectories, the A20 module is dispensable,

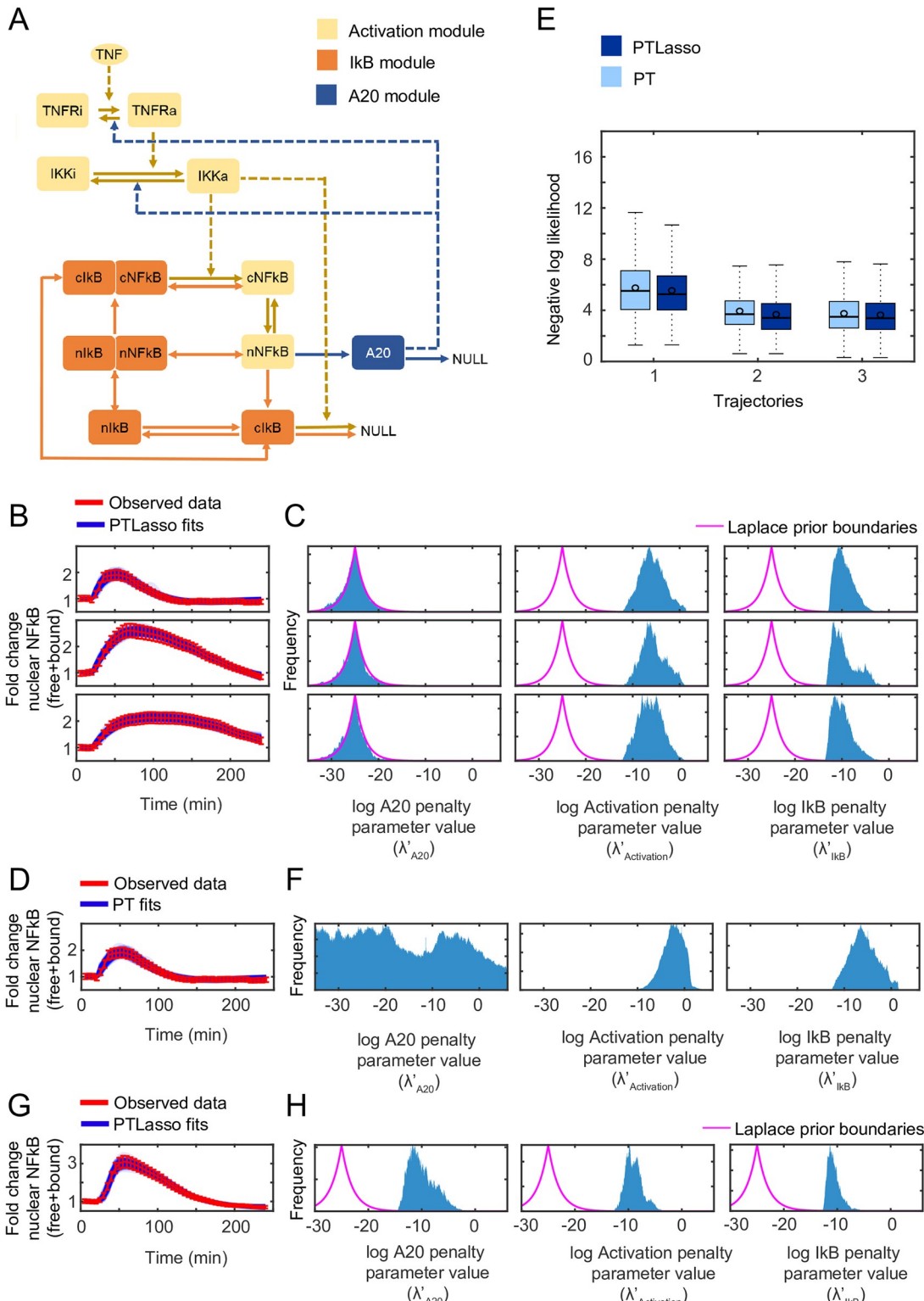

**Fig 4. Model reduction using grouped Lasso with a model of NF-κB signaling. A)** Reaction network diagram of a simplified model of TNF-NF-κB signaling. The colors indicate the different modules. Suffix "a" or "i" refer to active and inactive versions of the species respectively. Prefix "n" and "c" distinguish between nuclear and cytoplasmic versions of the species respectively. Solid lines indicate transformations and dashed lines indicate influences, where a species affects the rate of the corresponding reaction without being consumed. **B)** PTLasso fits of the model to three distinct single-cell NF-κB responses to pulsatile TNF stimulation.

**C)** Frequency histograms showing probability distributions of the penalty parameters (from 5,640,000 parameter samples) from model fits of single-cell NF-$\kappa$B responses (from panel B) to pulsatile TNF stimulation with PTLasso. **D)** PT fits of the model to the single-cell NF-$\kappa$B response from the first row of panel B to pulsatile TNF stimulation. **E)** Box plots comparing the log likelihood of the fits for NF-$\kappa$B responses to pulsatile TNF stimulation (from 5,640 parameter samples) with PT and PTLasso. Trajectories 1–3 correspond to the three trajectories in rows 1–3 of panel B. Boxes show data in the $25^{th}$–$75^{th}$ percentile and the circles show the mean. **F)** Frequency histograms showing probability distributions of the penalty parameters (from 5,640,000 parameter samples) from model fits of the single-cell NF-$\kappa$B response in panel D to pulsatile TNF stimulation with PT. **G)** PTLasso fits of the model to a single-cell NF-$\kappa$B response to continuous TNF stimulation. **H)** Frequency histograms showing probability distributions of the penalty parameters (from 3,200,000 parameter samples) from model fits of the single-cell NF-$\kappa$B response to continuous TNF stimulation with PTLasso. For panels B, D and G, transparent blue lines show ensemble fits (from 288 parameter samples for the fits with pulsatile TNF and 168 parameter samples for the fits with continuous TNF stimulation) of the model to single-cell NF-$\kappa$B responses. An NF-$\kappa$B response is calculated as the fold change of the sum of the abundances of bound and free NF-$\kappa$B in the nucleus. Red lines show the experimental data. Error bars show the 10% standard deviation assumed for the likelihood function during fitting and represent measurement error. Pulsatile TNF stimulation is 5 ng/ml for 5 minutes. Continuous TNF stimulation is 0.1 ng/ml. For panels C, F and H, the range of log parameter values on each x-axis covers the full range over which parameters were allowed to vary (−35 to 6 for pulsatile TNF stimulation and −30 to 6 for continuous TNF stimulation). The y-axis of each panel is scaled to the maximum value of the corresponding distribution to emphasize differences in shape. The pink lines show the boundaries of the Laplace prior with $\mu = -25$, $b = 2$. When fitting with PT, the penalty parameter distributions have uniform priors.

whereas the I$\kappa$B and activation modules are not. The A20 module might still be essential for other biology of the system, but the model does not require the A20 module to produce these single-cell NF-$\kappa$B responses under the given experimental condition and network constraints. The fits with PTLasso were as good as the fits with PT alone (Fig 4B and 4D), as is demonstrated by comparing the average log-likelihoods (Fig 4E), though the fits with PT produced broader distributions for the A20 parameters (Fig 4F, S6 Fig). The posterior distributions estimated for many of the reaction rate constants showed overlap with values previously reported in the literature [2, 39, 48] (S6 Fig).

To test the requirement of A20 feedback under different experimental conditions and network constraints, we also fit the model to a published single-cell NF-$\kappa$B response to continuous TNF stimulation [47] (Fig 4G). A soft constraint that IKK responses are transient was added for consistency with published observations [49, 50]. For responses to a TNF pulse, IKK activity naturally adapts back to its baseline abundance without additional negative regulation (S7 Fig). In this case, all three module penalty parameters deviate from the prior (Fig 4H), indicating that the A20 mediated negative regulation of IKK is essential for responses to continuous TNF stimulation. Taken together, the results for the NF-$\kappa$B signaling model provide an example where PTLasso isolates reaction modules sufficient for responses to specific experimental conditions and time scales.

## Discussion

In this work we have demonstrated that PT combined with Lasso is an effective approach to learn reduced models from a prior model with a larger number of reactions. Even when starting from a complete graph without prior knowledge of the underlying signaling network, PTLasso correctly identified reduced model architectures and reaction rate constants. PTLasso also correctly isolated subnetworks that are necessary for distinct dose-response relationships. In a model of NF-$\kappa$B signaling, PT with grouped Lasso found that in the absence of other network constraints, A20 feedback was not required to explain single-cell responses to a short TNF pulse, but is required when TNF treatment was continuous. Model reduction using PTLasso can therefore highlight aspects of the reaction network that are important for specific experimental conditions and timescales and not others.

Energy landscapes for systems biology models are often multimodal [51], which raises the possibility that multiple minimal models will fit the data. The fully connected five-node

network example demonstrates that PTLasso can identify multiple minima when present, but we also note that the posterior distributions for the parameters $k_{BC}$, $k_{BD}$, $k_{BE}$ were not identical as would be expected from symmetry (Fig 2D). These differences show that even when the PSRF and MPSRF values are below the standard thresholds, MCMC sampling methods may not obtain the correct probabilities for each possible solution. In a worst case, an apparently converged sample might miss a plausible mechanism.

Here, we have only used PTLasso to reduce ODE models of biochemical kinetics, but the method is in principle agnostic to the modeling formalism used. The grouped Lasso approach to select over reaction modules can even be adapted to select over abstract representations of signaling mechanisms, for example, coarse-grained nonlinear input-output functions, when detailed molecular reaction networks are not known. This may be useful to highlight pathways sufficient for certain experimental data in large multi-pathway models, such as whole-cell models [52] or models of signaling crosstalk [53], where it may not be possible or desirable to accurately represent each biochemical pathway in full mechanistic detail.

Another potential application of model reduction arises in fitting a model to data from different cell types. Differences in responses to the same experimental condition might be explained by differences in parameter values [33], but comparing cell-type specific parameter distributions in high dimensional space may be difficult when the models are non-identifiable. Reducing the number of model parameters lowers the dimensionality of the space and makes this problem easier.

A limitation of PTLasso is the large number of swaps required to reach convergence, which can lead to long execution times. For the simplest examples presented here, convergence happens on the order of hours on a standard workstation computer, but for the more complex signaling systems, convergence can take several days. Most of the execution time is dedicated to converging the joint parameter distribution. Currently PT and PTLasso are both run for fixed chain lengths followed by convergence testing at the end, often generating more samples than were required to pass convergence tests. Testing convergence on-the-fly and terminating the chains when convergence is reached would prevent unnecessary sampling and reduce the overall execution time. Approaches such as APT-MCMC [54] and Hessian-guided MCMC [22] that account for the shape of the parameter landscape during sampling could also reduce the number of samples required for convergence.

Along with working to reduce the amount of sampling, we are also investigating algorithmic modifications to reduce the execution time of individual PT swaps. Synchronous swapping in our current implementation of PT requires each chain to complete a fixed number of steps before attempting a swap. Because high temperature chains sample parameter space broadly and encounter regions where stiffness leads to long integration times, lower temperature chains often have to wait for the higher temperature chains to complete before swaps can be attempted. Asynchronous swapping [20] may therefore reduce execution times. Overall, there are still many opportunities for future PTLasso implementations to increase efficiency and applicability to larger systems biology models.

In this study we have presented a Bayesian framework that systematically dissects mechanistic ODE models of biochemical systems to identify minimal subsets of model reactions that are sufficient to explain experimental data. Technology now enables the building and simulating of highly detailed models that accurately reflect existing knowledge of a biochemical system. But detailed models may obscure our ability to identify underlying mechanisms. PTLasso serves as a bridge between these detailed models and simpler mechanistic explanations that are sufficient to account for system behavior under specific conditions.

## Supporting information

**S1 Fig. Hyperparameter tuning for PTLasso with a fully connected 3-node graph. A)** Data generated for fitting. Red dashed lines show the model simulation at 8 time points with the true parameter values. Each colored line represents a noisy trajectory obtained by adding Gaussian noise (mean = 0, standard deviation = 30% of the true data value) to the true data. The black error bars show the mean and standard deviation of the 10 repeats, and is the observed data used for fitting. **B)** Hyperparameter tuning plot showing variation in the negative log likelihood distribution (from 4,000 parameter samples) with $\mu$ and $b$ (red points show the mean, and black lines show mean ± standard deviation). The hyperparameters selected ($\mu = -10$, $b = 1$) provide the most regularization while not substantially increasing the negative log likelihood. **C)** Comparison of the log likelihood distributions (from 4,000 parameter samples) of the fits with PT and PTLasso ($\mu = -10$, $b = 1$). Box plots are obtained using a third party MATLAB library, aboxplot*, with outliers not shown. Boxes show data in the $25^{th}$–$75^{th}$ percentile and the circles show the mean. **D)** Example of PTLasso fits (from 4,000 parameter samples) where $b$ is too small ($\mu = -10$, $b = 0.1$) and the negative log likelihood of the fit is increased, and **E)** the corresponding parameter distributions (from 400,000 parameter samples). Since the regularization strength was too high, none of the parameters deviated from the prior. *http://alex.bikfalvi.com/research/advanced_matlab_boxplot/.
(TIF)

**S2 Fig. Hyperparameter tuning for PTLasso with a fully connected 5-node graph. A)** Data generated for fitting. Red dashed lines show the model simulation at 8 time points with the true parameter values. Each colored line represents a noisy trajectory obtained by adding Gaussian noise (mean = 0, standard deviation = 30% of the true data value) to the true data. The black error bars show the mean and standard deviation of the 10 repeats, and is the observed data used for fitting. **B)** Hyperparameter tuning plot showing variation in the negative log likelihood distribution with $\mu$ and $b$ (from 7,000 parameter samples, red points show the mean, and black lines show mean ± standard deviation). The hyperparameters selected ($\mu = -10$, $b = 1$) provide the most regularization while not substantially increasing the negative log likelihood. **C)** Box plots comparing the log likelihood distribution (from 7,000 parameter samples) obtained with PT and PTLasso for the chosen values of hyperparameters. Box plots are obtained using a third party MATLAB library, aboxplot*, with outliers not shown. Boxes show data in the $25^{th}$–$75^{th}$ percentile and the circles show the mean. **D)**. Parameter covariation of the three selected parameters with PTLasso and **E)** with PT shown as a 3D scatter plot with transparent points (from 700,000 parameter samples). *http://alex.bikfalvi.com/research/advanced_matlab_boxplot/.
(TIFF)

**S3 Fig. Model reduction using PTLasso with fully connected 3-node and 5-node graphs when the observed data is generated from noisy parameters. A)** Noisy parameter values (black) used to generate the observed data. The log true parameters (red) of the known model were perturbed 10 times with Gaussian noise (mean = 0, standard deviation = 0.05). **B)** Colored lines show model outputs for each of the 10 noisy parameter sets. The black error bars shows the mean and standard deviation of the colored lines and is the observed data for fitting. Red dashed line shows the model simulation at 8 time points with the true parameter values. **C)** Frequency histograms showing probability distributions of the parameters (from 800,000 parameter samples) for PTLasso fits of a fully connected three node graph and **D)** fully connected five node graph. The range of log parameter values on each x-axis is −12 to 3, which covers the full range over which parameters were allowed to vary. The y-axis of each panel is

scaled to the maximum value of the corresponding distribution to emphasize differences in shape. The pink lines show the boundaries of the Laplace prior with $\mu = -10$, $b = 1$, and the dashed red lines in panels for $k_{AB}$ and $k_{BC}$ show the true parameter values. A parameter distribution confined within the Laplace prior boundaries indicates that the parameter is extraneous. **E)** PTLasso fits to the data for a fully connected three node graph and **F)** five node graph. Transparent blue lines show ensemble fits (from 8,000 parameter samples, 100 time points per trajectory), red line shows the true data (100 time points), and the black error bars show the mean ± standard deviation of the observed data (8 time points).
(TIF)

**S4 Fig. Hyperparameter tuning for PTLasso with dose-response motifs inferred from a prior network. A)** Linear correlation of non identifiable parameters in the reduced perfectly adapting model shown as a scatter plot (axes show log parameter values). **B)** Hyperparameter tuning plot for the linear dose response model and **C)** the perfectly adapting dose response model. The hyperparameter tuning plot shows variation in the negative log likelihood distribution with $\mu$ and $b$ (from 400 parameter samples for the linear dose response model and from 800 parameter samples for the perfectly adapting dose response model. Red points show the mean, and black lines show mean ± standard deviation). The hyperparameters selected ($\mu = -10$, $b = 0.5$ for linear dose-response and $\mu = -10$, $b = 1$ for perfectly adapting dose-response) provide the most regularization while not substantially increasing the negative log likelihood. **D)** Box plots comparing the log likelihood distribution obtained with PT and PTLasso for the chosen values of hyperparameters for the linear dose response model (from 400 parameter samples) and **E)** the perfectly adapting dose response model (from 800 parameter samples). Box plots are obtained using a third party MATLAB library, aboxplot*, with outliers not shown. Boxes show data in the $25^{th}$–$75^{th}$ percentile and the circles show the mean. *http://alex. bikfalvi.com/research/advanced_matlab_boxplot/.
(TIF)

**S5 Fig. Hard constraints on parameter covariation in NF-κB signaling.** Binned scatter plots (MATLAB function binscatter with 940,000 parameter samples from a PTLasso fit for an NF-κB response to pulsatile TNF stimulation) show the joint distributions for the pairs of parameters for which covariance was constrained during fitting.
(TIF)

**S6 Fig. Posterior probability distributions of model parameters shown with the corresponding published values for a representative NF-κB response to pulsatile TNF stimulation. A)** with PTLasso and **B)** with PT. Distributions of total protein abundance parameters and rate constant parameters (from 5,640,000 parameter samples) from the A20 module (blue), Activation module (yellow) and IκB module (orange). All parameters are in logscale. Total protein abundance parameters have uniform priors, and the x-axis range indicates the sampling range. Rate constant parameters are sums of the module penalty parameters and reaction-specific parameters. The pink line corresponds to a best-fit parameter set. The dashed lines correspond to published values of parameters—Pekalski et al. [2] (red), Lee et al. [39] (black), and Kearns et al. [48] (blue). A published parameter value for a model is only included if the corresponding reaction maintained the same structure in both the published and current models. For unit conversions we used the values mentioned in Lee et al. [39], $1\mu M$ of NF-κB = 50,000 molecules/cell and applied this to other species in the models. In the parameter labels "import" refers to translocation from the cytoplasm into the nucleus and "export" is the reverse. "Complex" refers to the NF-κB-IκB complex. The y-axis for each panel is scaled from

0 to the maximum value of the distribution to emphasize differences in the shapes of the distributions.
(TIF)

**S7 Fig. NF-$\kappa$B signaling model predictions.** Model predictions for non-fitted variables for a representative NF-$\kappa$B response to pulsatile TNF stimulation (from 500 parameter samples from one of the PTLasso repeats). Suffix "a" and "i" refer to active and inactive versions of a species respectively. Prefix "c" and "n" refer to cytoplasmic and nuclear versions of a species respectively. "Complex" refers to the NF-$\kappa$B-I$\kappa$B complex. Time courses are shown for 4 hours after the initial 5ng/ml TNF stimulation. The TNF concentration is set to 0 at the 5 minute time point.
(TIF)

**S1 Table. Hard constraints in the NF-$\kappa$B signaling fits.**
(PDF)

**S2 Table. Maximum PSRF across all model parameters for each example shown up to 4 significant digits.** Parameter distributions are constructed from the lowest temperature chain.
(PDF)

**S3 Table. MPSRF values for parameter distributions from each example shown up to 4 significant digits.** Parameter distributions are constructed from the lowest temperature chain.
(PDF)

**S4 Table. PSRF to show convergence of energy distributions when combining PT or PTLasso chains for the NF-$\kappa$B signaling fit with pulsatile TNF stimulation.** M is the number of independent chains that are combined for each group and N is the total number of swaps. The length of each chain is N/M. Energy distributions are constructed from the lowest temperature chain.
(PDF)

**S5 Table. PSRF to show convergence of energy distributions when combining PTLasso chains for the NF-$\kappa$B signaling fit with continuous TNF stimulation.** M is the number of independent chains that are combined for each group and N is the total number of swaps. The length of each chain is N/M. Energy distributions are constructed from the lowest temperature chain.
(PDF)

**S6 Table. Step acceptance rates for the lowest temperature chain for each example.**
(PDF)

**S7 Table. Swap acceptance rates for the two lowest temperature chains for each example.**
(PDF)

**S8 Table. Reactions in fully connected three node network.** The "parameters" column specifies the forward and reverse rate constant pair. True parameters are shown in red. All the reactions follow mass action kinetics. First order reaction rate constants are in units of s$^{-1}$. Second order reaction rate constants are in units of molecule$^{-1}$s$^{-1}$.
(PDF)

**S9 Table. Reactions in fully connected five node network.** The "parameters" column specifies the forward and reverse rate constant pair. True parameters are shown in red, and parameters of the inferred alternate reduced models are shown in blue. All the reactions follow mass

action kinetics. First order reaction rate constants are in units of $s^{-1}$. Second order reaction rate constants are in units of $molecule^{-1}s^{-1}$.
(PDF)

**S10 Table. Prior network comprising linear dose-response model reactions and adaptive dose-response model reactions.** The "parameters" column specifies the forward and reverse rate constant pair. All the reactions follow mass action kinetics. First order reaction rate constants are in units of $s^{-1}$. Second order reaction rate constants are in units of $molecule^{-1}s^{-1}$.
(PDF)

**S11 Table. Reactions in NF-$\kappa$B signaling model.** The "parameters" column specifies the forward and reverse rate constant pair. All the reactions follow mass action kinetics. First order reaction rate constants are in units of $s^{-1}$. Second order reaction rate constants are in units of $molecule^{-1}s^{-1}$ except $k_b$ which is in units of $(ng/ml)^{-1}s^{-1}$.
(PDF)

## Acknowledgments

We thank all the members of the Lee and Faeder laboratories for many helpful discussions.

## Author Contributions

**Conceptualization:** Sanjana Gupta, Robin E. C. Lee, James R. Faeder.

**Formal analysis:** Sanjana Gupta.

**Funding acquisition:** Robin E. C. Lee, James R. Faeder.

**Investigation:** Sanjana Gupta, Robin E. C. Lee, James R. Faeder.

**Methodology:** Sanjana Gupta, James R. Faeder.

**Project administration:** Robin E. C. Lee, James R. Faeder.

**Software:** Sanjana Gupta, James R. Faeder.

**Supervision:** Robin E. C. Lee, James R. Faeder.

**Validation:** Sanjana Gupta.

**Visualization:** Sanjana Gupta, Robin E. C. Lee, James R. Faeder.

**Writing – original draft:** Sanjana Gupta.

**Writing – review & editing:** Sanjana Gupta, Robin E. C. Lee, James R. Faeder.

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
