## [Decision Letter · Decision Letter 0]

20 Nov 2019

Dear Dr Faeder,

Thank you very much for submitting your manuscript 'Parallel Tempering with Lasso for Model Reduction in Systems Biology' for review by PLOS Computational Biology. Your manuscript has been fully evaluated by the PLOS Computational Biology editorial team and in this case also by independent peer reviewers.

The reviewers appreciated the attention to an important problem, but raised several concerns about the manuscript as it currently stands. In particular, the reviewers have questions about details related to the mathematical approach for parameter fitting, including convergence criteria and chain swapping. In addition, more attention should be paid to presentation of the figures, parameter values, and which data to include in the main text compared to supplement.

While your manuscript cannot be accepted in its present form, we are willing to consider a revised version in which the issues raised by the reviewers have been adequately addressed. We cannot, of course, promise publication at that time.

Sincerely,

Stacey Finley, Ph.D.

Associate Editor

PLOS Computational Biology

Thomas Lengauer

Methods Editor

PLOS Computational Biology

[LINK]

Reviewer's Responses to Questions

**Comments to the Authors:**

Reviewer #1: In this manuscript, Gupta, Lee, and Faeder describe a method for identifying key kinetic modules associated with more complicated mechanistic models using a Bayesian parameter estimation approach coupled with Lasso regularization to reduce the number of parameters. In terms of the review criteria:

Originality - yes, the work appears to be original as it incorporates Lasso regularization. Of note, Klinke and Finley used a Bayesian parameter estimation approach coupled with time-scale separation for model reduction (Klinke and Finley Biotech Prog 2012).

Significant biological and/or methodological insight - methods that enable one to extract some biological insight from a more complicated mechanistic model are helpful. This one uses Lasso regularization as a new aspect.

Rigorous methodology -

- Couple of concerns:

1. The authors choose to incorporate a Lasso regularization approach to minimize the number of parameters in the model. Given that parameters in system biology models may not be tightly constrained and may exhibit correlation with other parameters, a Lasso approach only chooses one parameter in a set of correlated parameters. The choice is a bit arbitrary. So what is it to say if the Lasso chooses another parameter in the correlated set, would the conclusions be different?

2. Given the common presence of parameter non-identifiability in these types of models, convergence applied to the parameters doesn't really make sense, while applied to the model predictions does. Once the chains have converged, they are sampling the posterior distribution. It seems here, the authors only ran the MCMC chains until they converged and then used the unconverged segments. That's not correct.

3. It's unclear to me about the point of swapping chains. In this implementation, the proposed step only depends on the current point. Swapping chains just artificially inflates within chain variance and reduces between chain variance such that the Gelman-Rubin PSRF would be reduced, suggesting early convergence. Of note, the majority of the time is spent solving ODEs - so figure out a way to speed up solving the ODEs, especially stiff ones.

Importance to researchers in the field - Yes, this may be helpful to researchers in the field to reduce the complexity of models based on the data available.

Other minor comments:

Unclear how PT is better than adaptive MCMC methods, which are much faster than plain MCMC.

Unclear how noise properties impacts results using toy model. Experimentally, noise distribution may be constant (i.e., independent of the true value) or may be inversely proportional to the true value (RNAseq data is like that).

Reviewer #2: This paper presents a novel computational method (PTLasso) for identifying reduced mathematical models of signaling networks. By searching for reduced models that can fit temporal experimental data, the method yields information on critical pathway components, modules, or parameters that are necessary to explain observed data. In my opinion, this paper addresses major needs in the area of quantitative systems biology/mathematical biology: first, by seeking reduced models it addresses the issue of overfitting (which is a particular challenge in cell signaling, where models based on prior knowledge are becoming increasingly complex, even while very limited information about parameter values is available); second, it addresses the issue of how to interpret model fits (by revealing which aspects of a model are actually necessary to reproduce experimental observations). Overall I believe this paper makes a significant contribution to cell signaling and other areas of quantitative biology, which should be published and is a good fit for PLOS Comp. Bio. The methodological approach is solid.

However, there are a number of issues with the presentation that should be addressed before publication, detailed below.

1) Throughout the paper, the mathematical models are presented only graphically. If I understand correctly, all models are treated as ODEs with added Gaussian noise. I think it would be helpful to include in the Supplement the explicit equations for all models. This would aid reproducibility and help prevent potential confusion, for example:

a) I was initially confused about the usage of both capital Ks and lower-case ks to denote kinetic rate parameters, since capital Ks are typically associated with equilibrium constants (i.e., ratios of kinetic parameters). I would recommend using only lower-case ks (e.g., in Figures 1 and 2), but regardless, showing the full models would clarify any potential confusion about what the parameters mean.

b) In the caption of Figure 3 model description, it reads “Solid lines show species conversions and dashed lines show influences”. I think I know what is meant by this in terms of mass-action kinetics, but I’m not sure it would be universally understood. Again, the full equations would prevent any confusion.

2) In the captions of Figs. 1-3, the parameter values are reported without any units. The units should be clarified (I assume they are per-second rate constants, since the simulated time trajectories are presented in seconds)

3) It would be helpful if the x-axis could be labeled with numeric values for the parameter plots (e.g., Fig 1C, Fig 2C and D, etc.). With the plots unlabeled, it’s very difficult to figure out where the prior was centered (the value of mu) and how that value compares to the true parameter values. I realize this is a challenge because of space limitations. At least, it would be helpful to clarify the chosen values of mu and b directly in the figure and/or figure caption, along with the values of the true parameters for comparison.

4) Throughout the paper, it is unclear what is meant by the number of samples. For example, in Figure 1B, the curves represent “4e3 samples” and in Figure 1C, the distributions show “4e5 PT samples” (is it the same number for both PT and PTLasso?). Why is the number of samples shown in 1B and 1C different? I had a hard time understanding (1) what does one sample mean, in terms of the MCMC procedure described. Does one sample = one run of many iterations? If so, is it subject to the convergence criterion described in Methods? (2) What determines the number of samples used, which is different for different models?

5) The results for NFkB signaling seem central to the paper, since the authors here show an application to real-world experimental data. I think the presentation of Figure 4 and corresponding section of Results could be improved:

a) It’s unclear to me why Fig. S7 is in the supplement, and not included in the main text (perhaps as part of Figure 4, or as a stand-alone figure). It seems like an important result that demonstrates the utility of the method: that PTLasso identifies the A20 network module as being necessary to reproduce the nNFkB trajectories under continuous stimulation, whereas it is NOT necessary to reproduce measurements from short-pulse stimulated cells. From a biological standpoint, it suggests that cells may engage negative feedback loops (or other types of modules) only under certain stimulus conditions. This is very interesting!

b) The presentation in Fig. 4 is confusing, because in 4C results for both PT and PTLasso are shown, whereas in 4B only “PT Fits” are mentioned, but I believe PTLasso is implied.

c) In Fig. 4, only the PTLasso penalty parameter distributions are shown, which seems inconsistent in presentation with the rest of the paper, in which distributions of parameters from both PT and PTLasso are directly compared. (And since PT serves as the baseline of comparison throughout the paper, to support the utility of PTLasso). I guess the challenge is that there are so many parameters in the NFkB signaling network. Still, it would be nice to see a visual representation of the output of PT versus PTLasso, especially since it is emphasized that the PT and PTLasso fits are equally good (4C). I was particularly interested in whether the PT fits also hint at some modularity, or do the PT parameters tend to involve all modules? In other words, is PTLasso really necessary to reach the conclusion that the A20 module is not necessary, or can PT alone reach this conclusion, perhaps indirectly? I wonder if there’s a way to show parameter ensembles, somehow lumped by module. Another solution would be to directly show the PT and PTLasso parameter distributions in the Supplement, where space is not limiting (and group the parameters by module). I think this is already done for PTLasso in Fig. S5, where the A20 module parameters tend to be centered around low values (-20 to -30). Do the corresponding plots for PT show higher values and/or broader distributions of A20-module parameters? This would support the idea that PTLasso is really necessary over PT to eliminate the A20 module.

d) I don’t think Fig. S5 is ever referred to in the Main Text (I couldn’t find it). If I understand, it shows that PTLasso recovers best fit parameter values that are in line with previous estimates. This seems worth describing in Main Text. However, it needs to be clarified which NFkB trajectory these parameters correspond to: continuous or short-pulse stimulation.

6) This might be a minor detail, but in line 144, the authors neglect the normalization constant 1/2b. If I’m not mistaken, this means that the weight of the prior (relative to the likelihood) in the energy function is changing in a way that depends on the hyperparameter b. Is this justified?

7) There are some typos in line 113: missing vector notation on theta, and a missing period.

8) Could the authors clarify the relationship between equations 1 and 2? Is there an explicit equality between p(Y given theta) and L(theta)?

Reviewer #3: The paper by Gupta et al addresses an important problem in computational biology - the calibration of parameters in ODE kinetic models and model reduction. The authors extended their past work on parallel tampering and Bayesian MCMC chains to include Lasso regularization. Regularization is very important in large models and the Lasso framework they use merges very well with the parallel tempering algorithm. Overall I recommend that the paper be published by Plos Comp Bio. I have a few suggestions that I think can improve the paper without too much effort and I recommend that these will be suggested to the authors so they can decide which to implement.

I did not understand how the authors dealt with parameter units. Parameters have values that have are on an arbitrary scale, some could be order of magnitude different from others. In their MCMC runs, parameter scaling matters in their multivariate proposal distribution ( eq 4.). This is especially true if parameter covariance is taken into account. One suggestion (see Yao et al MSB 2016) is to normalize all values related to a reference value using log of the fold difference. There are other ways of addressing this issue and perhaps the authors already do and I missed it. But if they don’t I think that they should.

The authors used synthetic data to test their methodology. I think this is a great approach. However, the way they implemented the addition of noise in synthetic data is problematic. The addition of Gaussian white noise is very unrealistic. The majority of variability in biological measurement is not due to instruments where the idea of Gaussian noise is reasonable but from biological variability. An alternative I would suggest is to add parameter variability (that could be gaussian) and run multiple simulations of the ODEs and use that to generate the synthetic data and see how good is model calibration is under these conditions.

In the NFkB model, the authors used both hard and soft constraints. It will be great to know if these were essential, i.e. can the system “work”, create good fits even if these constraints are violated?

The authors explain how they tune hyperparameters related to the prior. But I did not see any discussion related to other hyperparameters related to the proposal distribution, the tempering rate, the likelihood function (i.e. the sigma of the residual). Are these less important? Or should they also be tuned?

**Have all data underlying the figures and results presented in the manuscript been provided?**

Reviewer #1: Yes

Reviewer #2: Yes

Reviewer #3: None

PLOS authors have the option to publish the peer review history of their article (what does this mean?). If published, this will include your full peer review and any attached files.

Reviewer #1: No

Reviewer #2: No

Reviewer #3: No

---

## [Decision Letter · Decision Letter 1]

20 Jan 2020

Dear Dr. Faeder,

We are pleased to inform you that your manuscript 'Parallel Tempering with Lasso for Model Reduction in Systems Biology' has been provisionally accepted for publication in PLOS Computational Biology.

Before your manuscript can be formally accepted you will need to complete some formatting changes, which you will receive in a follow up email. A member of our team will be in touch within two working days with a set of requests.

Best regards,

Stacey Finley, Ph.D.

Associate Editor

PLOS Computational Biology

Thomas Lengauer

Methods Editor

PLOS Computational Biology

Reviewer's Responses to Questions

**Comments to the Authors:**

Reviewer #1: The revision seems to largely address the prior critiques.

Reviewer #2: The authors have addressed all my concerns. I think this is a great paper and it is ready to be published.

Reviewer #3: The authors addresses all my concerns. I recommend the paper for immediate publication in Plos Comp Bio

**Have all data underlying the figures and results presented in the manuscript been provided?**

Reviewer #1: Yes

Reviewer #2: None

Reviewer #3: Yes

PLOS authors have the option to publish the peer review history of their article (what does this mean?). If published, this will include your full peer review and any attached files.

Reviewer #1: No

Reviewer #2: No

Reviewer #3: No

---

## [Editor Report · Acceptance letter]

28 Feb 2020

PCOMPBIOL-D-19-01641R1 

Parallel Tempering with Lasso for Model Reduction in Systems Biology

Dear Dr Faeder,

I am pleased to inform you that your manuscript has been formally accepted for publication in PLOS Computational Biology. Your manuscript is now with our production department and you will be notified of the publication date in due course.

With kind regards,

Laura Mallard
